# A National Accessibility Audit of Primary Health Care Facilities in Brazil—Are People with Disabilities Being Denied Their Right to Health?

**DOI:** 10.3390/ijerph18062953

**Published:** 2021-03-13

**Authors:** Alexandro Pinto, Luciana Sepúlveda Köptcke, Renata David, Hannah Kuper

**Affiliations:** 1Oswaldo Cruz Foundation, Brasília 70904-130, DF, Brazil; alexandro.pinto@fiocruz.br (A.P.); luciana.koptcke@fiocruz.br (L.S.K.); renata.david@fiocruz.br (R.D.); 2International Centre for Evidence in Disability, Department of Clinical Research, London School of Hygiene & Tropical Medicine, London WC1E 7HT, UK

**Keywords:** accessibility, barriers, disability, Brazil, primary care, health facilities

## Abstract

Poor accessibility of healthcare facilities is a major barrier for people with disabilities when seeking care. Yet, accessibility is rarely routinely audited. This study reports findings from the first national assessment of the accessibility of primary health care facilities, undertaken in Brazil. A national accessibility audit was conducted by trained staff of all 38,812 primary healthcare facilities in Brazil in 2012, using a 22-item structured questionnaire. An overall accessibility score was created (22 items), and three sub-scales: external accessibility (eight items), internal accessibility (eight items), information accessibility (six items). The main finding is that the overall accessibility score of primary care facilities in Brazil was low (mean of 22, standard deviation (SD) of 0.21, on a 0–100 scale). Accessibility of different aspects of the healthcare facilities was also low, including external space (mean = 31.0, SD = 2.0), internal space (18.9, 1.9) and accessibility features for people with other visual or hearing impairments (6.3, SD = 1.0). Scores were consistently better in the least poor regions of Brazil and in facilities in larger municipality size (indicating more urban areas). In conclusion, large-scale accessibility audits are feasible to undertake. Poor accessibility means that people with disabilities will experience difficulties in accessing healthcare, and this is a violation of their rights according to international and Brazilian laws.

## 1. Introduction

Globally, there are approximately one billion disabled people, equating to one in seven people worldwide [1]. People with disabilities, on average, have higher healthcare needs than their peers without disabilities [1]. They may require specialist services, such as ophthalmology, physiotherapy or mental health care as a result of their impairment. Disabled people may also have greater needs for general healthcare services as on average they are more likely to experience poor health [2], because they are generally older and poorer, and the health condition and impairment that underlies their disability may have further health consequences (e.g., pressure sores among people with spinal cord injuries). Overall, therefore, disabled people are more likely to need healthcare services, whether general or specialised, although the entry point to both is usually through access to primary healthcare [3,4]. At the same time, people with disabilities frequently experience a range of barriers which make accessing healthcare difficult [1,5]. These barriers include lack of physical accessibility of healthcare facilities, lack of affordability of care, and lack of availability of services needed, as well as issues surrounding quality of care received [3,4]. Consequently, disabled people may experience worse access to healthcare services [3,4], which is a violation of their rights, as established in the UN Convention on the Rights of Persons with Disabilities (UNCRPD) and by the laws of most countries [6].

Solutions are therefore needed to improve healthcare access for people with disabilities, and these must be targeted at overcoming the barriers faced. Improving accessibility of facilities is clearly a key need [1] and is specifically highlighted within the UNCRPD (Article 9) [6]. Healthcare facilities must be physically accessible, which means that people with disabilities are able to enter the building, move around inside, and use the bathrooms. Clinical equipment must be useable by people with different impairments–for instance–adjustable beds should be available for people with mobility impairments. But, provision of ramps and so on are not the only issue. People with visual impairments may need braille signage in order to move independently. And accessibility of information is also important–for instance for people with hearing loss or intellectual impairments may need access to services such as a hearing loop, or for information to be provided in simplified formats.

Many countries have laws or policies in place that mandate that certain access standards are met in healthcare provision (e.g., Americans with Disabilities Act 1990, UK Equality Act 2010, Brazilian law on the Inclusion of Persons with Disabilities 2015). These rules are ineffective if they are not monitored. Accessibility audits can be used for monitoring purposes to understand whether facilities are adhering to certain standards, and a wide variety of tools are available (e.g., Americans with Disabilities Act Checklist, Health Facilities Scotland-Access Audit, National Disability Authority Ireland Access-Handbook). Many of these tools are complex, requiring measurement of slopes of ramps and widths of doors, and so on. As a consequence, few large-scale accessibility audits have been conducted, and almost all available data on accessibility of healthcare facilities is from high income settings and focusses on physical accessibility alone [7,8,9]. Data are lacking from low and middle-income countries (LMICs) [10,11], and the vast majority of data available is from Brazil [12,13,14,15,16,17]. Existing assessments from Brazil show that there are important issues in accessibility of healthcare facilities [13,14,15]. However, many of these studies are small [14,15], or conducted in specific locations (e.g., Ceara state) [14,15,16], and so cannot be generalized across the entirety of Brazil. Moreover, there is likely to be widespread variation, as Brazil is an enormous country, and factors such as municipality size, facility type and region of Brazil will likely influence the accessibility of clinics.

Fortunately, there is also a strong commitment not only to accessibility of healthcare services, but also to data collection and accountability in Brazil, as a result of the efforts by the Ministry of Health (MoH) in recent decades. The MoH has created a national monitoring framework for healthcare, which includes assessment of accessibility of facilities [18,19,20]. The National Program for Access and Quality Improvement in Primary Care (PMAQ-AB) is an initiative of the MoH for the implementation of the Primary Health Care (PHC) evaluation. It was introduced in 2011, under the scope of the National Primary Care Policy (PNAB). This study aims to present a national assessment of the accessibility of primary health care facilities for people with disabilities in Brazil, using data collected routinely from the public health system in the first cycle (2012) PMAQ-AB. The research questions are: (1) What is the accessibility of primary health care facilities in Brazil (overall, internal, external, information), and (2) How is accessibility related to region, municipality size, and characteristics of the facility (e.g., number of healthcare workers, services offered, poverty level of region).

## 2. Materials and Methods

### 2.1. Study Design

This study used secondary data obtained from the External Evaluation of the Brazilian National Program for Improving Access to Primary Care–PMAQ-AB. The first cycle of the PMAQ-AB was carried out in 2012 across 5543 cities (99% of Brazilian municipalities). It included the completion of a census of the structure and organization of PHC services [21,22]. This survey was repeated in 2015 and 2018, but only on a sub-set of facilities. For these analyses, we used survey data from 2012, because it represents a complete national picture.

### 2.2. Data Collection

Between May–October 2012, PMAQ-AB made a census of the health facilities infrastructure of 38,812 public PHC facilities in Brazil, where the PHC was functional and the PHC manager agreed to participate in PMAQ-AB. A public primary health facility is defined by Brazilian Ministry of Health as a public building or mobile space, where an individual or a legal entity performs primary health care actions and services, and which has a technical, personnel and infrastructure compatible with its purpose.

The audit was undertaken by experienced researchers from 45 universities, hired by the Brazilian Ministry of Health. The researchers received a 20 h-long training by the universities, with a focus on the questionnaires, the possible difficulties that could arise during fieldwork and how to approach the research participants.

The researchers visited each public primary healthcare facility, and undertook an audit which included investigation of the building accessibility (internal and external) and the availability of healthcare personnel trained in disability-related issues. This assessment was undertaken with a 22 item (yes/no) electronic questionnaire and was completed through on-site observation (Table 1).

### 2.3. Data Processing and Analysis

By default, the data were transmitted directly online to a central dataset in the collaborating universities. When there was no internet access in the field, the data were stored on the electronic devices and transmitted to a central dataset at a later date. The data were obtained from the MoH by request in November 2019. Due to the MoH webpage overhauling, the data were not available freely in the internet.

We used a 22-item questionnaire to develop a total accessibility score (Appendix A). For each positive answer, one point was added to the score. The questions were subdivided in three sub-scales: external accessibility (8 items), internal accessibility (8 items), information accessibility (6 items). The scores were created by summing the relevant items and then transforming them into a 0–100 score. The Overall Accessibility Score was calculated by the sum of the three sub-scales and transformed into a total ranging between 0–100. Each sub-scale was attributed the same weight, regardless of the number of questions in the sub-scale, so that each dimension of accessibility was valued equally. There was missing data for one facility, and this was deleted in the dataset.

A priori we hypothesized that primary health care facilities in urban areas, richer regions and with a higher number of professionals would have better accessibility. We therefore took account of these factors in the analyses. The total number of professionals employed in the units (work force) was estimated by the sum of the professionals (physicians, dentists, nurses, etc.) registered to the units. The number of professionals was not known for some facilities (2.18% 1 (except for the number of “other professionals”, the percentage of the answer “don’t know” was less than 1%)), in which case we attributed the mean of the number of the specific health professional, considering the size and region of the municipality. The municipalities were classified according to their size into major (>900,000 inhabitants), large (100,001–900,000 inhabitants), medium (100,000–50,001 inhabitants) and small cities (≤50,000 inhabitants). Regions were ranked by poverty level based on Gross Regional Product (in USD billion in 2016): North (97.2), Central-West (175.7); Northeast (257.1); South (305.5), Southeast (981.4). [23]

Data analysis was conducted by cross-tabulating accessibility outcomes with key characteristics of the primary health care facilities (Region of Brazil, Municipality size, Type of facility). The statistical significance of the difference between the proportions was calculated by Tukey’s ‘Honest Significant Difference’ method, using a cut-off of a *p*-value less than 0.05. We used pseudo-Poisson regression models with robust confidence intervals in order to calculate the magnitude of relationship between the Overall Accessibility Score and the aforementioned characteristics [24]. Crude and adjusted analyses was performed. Statistical analysis was undertaken in R statistical program.

The regression equation is as follows:E[PR | PRR1, PRR2, PRR3] = exp (b0) × PRR1 × exp (Region) × PRR2 × exp (Municipality Size) × PRR3 × exp(Type)
where: E [PR | PRR1, PRR2, PRR3]: expected value for overall accessibility given PRR1, PRR2, PRR3, PR: Overall Accessibility Ratio, b0: intercept, PRR1: Prevalence ratio for Region, PRR2: Prevalence ratio for municipality size and PRR3: Prevalence ratio for Primary Health Unit type

### 2.4. Ethics

Submission to a research ethics committee was not required for this secondary analysis of openly-accessible data. Furthermore, data was not included on individuals and was not of a sensitive nature.

## 3. Results

In total, in 2012 there were 38,812 primary health facilities spread over more than 99% of the municipalities in the five regions of Brazil (Table 2). The proportion of facilities in the five regions closely follows the population distribution, with some under-representation of primary health facilities in the Southeast region (31% of facilities for 42% of the population) and over-representation in the Northeast (38% of facilities for 28% of the population). The primary health facility workforce distribution followed a similar pattern. Overall, the vast majority of facilities were based in small city sizes, and most were staffed with “graduated professionals” (i.e., physicians, nurses and/or dentists).

The accessibility of the external area of primary health care facilities was generally low and varied by region and municipality size (Table 3). Fewer than half of the facilities had appropriate flooring at the entrance (i.e., non-slip-28%, regular-49%, or smooth-36%), or an access ramp (44%). Few facilities had a handrail at the entrance (8%) and only one in three had an accessible entrance (35%). These measures of access were consistently better in the Southeast and South regions of Brazil, which were the least poor, compared to the other regions. The facilities in larger municipality size also had better indicators of access than those in smaller municipalities (indicating more rural areas). The overall score for external space accessibility reflected these trends; Scores were low overall (mean of 31.0 on a 0–100 scale) and were consistently higher for the wealthier regions of Brazil and the facilities in larger municipalities.

The patterns for accessibility of the internal space were similar to those of external spaces (Table 4). Accessibility of internal areas was low. For instance, only 12% of facilities had an adapted restroom, 24% had corridors that were wheelchair accessible and only 9% of facilities had adapted drinking fountains. Consistently, the indicators for internal accessibility were best for the richest Southeast and South regions, and worst for the poorer Northeast and North regions. Furthermore, facilities in larger municipalities had better internal access than those in smaller municipalities. The overall internal space accessibility score was low and was approximately twice as high in the Southeast/South regions (25.2–26.4) or largest municipality sizes (32.4) compared to the Northeast/North regions (9.5–12.1) or smallest municipalities (17.2).

Accessibility features of primary healthcare facilities for people with visual or hearing impairments were worst of all (Table 5). Almost none of the facilities displayed international symbols, had signs in embossed characters or braille or had hearing resources available. Just 9% of facilities showed signage that would help people with hearing impairments. Only one in five facilities (21%) had a professional trained in supporting people with disabilities. The mean overall score for accessibility in this category was only 6.3, compared to a possible total score of 100. Again, these measures were consistently better in richer, compared to poorer, regions, and in facilities in the largest, compared to smallest, municipalities.

Table 6 shows the magnitude of the association between the Overall Accessibility Score (OAS) and the public PHC facilities features: region, municipality size and kind of services rendered. The higher the ratio, the better is the accessibility compared to the baseline, for example, the South region OAS is twice as high (2.0) as the North region (baseline). The overall OAS in Brazil was 18.7 (SD 1.3), showing that accessibility is on average far below the possible 100 point level. Accessibility was consistently worse in the poorer regions, with scores approximately two times higher in the least poor compared to the poorest regions. Similarly, accessibility scores were higher in the larger municipalities and in those that had specialist workforce, compared to those that did not. These associations were changed little after adjustment by other variables.

## 4. Discussion

Brazil has made a great commitment to the achievement of Universal Health Coverage, through the establishment in 1989 of Sistema Único de Saúde (SUS), a national publicly funded healthcare system [25]. The Brazilian Federal Constitution of 1988, established an innovative and complex federal organization focusing on political and administrative decentralization, with direct consequences for health policies. Emphasis was put on the municipality to implement and deliver primary health care services to the population [26]. Furthermore, the Brazilian Constitution (1988) and a range of laws also protect the right to healthcare among people with disabilities. For instance, the National Policy for the Inclusion of Persons with Disabilities (1989, 1999) guarantees access of people with disabilities to a wide range of healthcare services, and the Brazilian Law for Inclusion of People with Disabilities (2015) reinforces the rights of people with disabilities for priority access to healthcare and rehabilitation. These laws are also reflected in policies, such as the National Policy on the Health of Persons with Disabilities (2002). Nevertheless, the findings from the 2012 national audit of 38,812 primary health facilities shows that physical accessibility of primary health care facilities was generally low, whether assessed in terms of the accessibility of the external area or internal space. Accessibility of primary healthcare facilities for people with visual or hearing impairments was even worse, with few making adaptations for these groups. Worryingly, only one in five facilities (21%) had a professional trained in supporting people with disabilities. Accessibility measures were consistently better in the richer Regions (South and Southeast) compared to the poorer parts of Brazil. Accessibility was also generally better in facilities in larger municipalities, indicating that they were in urban rather than rural areas. The worse accessibility of PHC facilities in small municipalities highlighted in our study may relate to fact that they are more dependent on federal incentives, and so their success in providing the service depends on their political assets, as well as their administrative and technical capacities. Attempts to overcome structural inequalities and their impact on the health system have been made by the government, but proved insufficient so far [27].

Existing assessments of accessibility of healthcare facilities in Brazil are consistent with our findings in showing that there are important issues facing people with disabilities. According to Albuquerque et al., analysing data from the Pernambuco state PMAq, only one among 2019 facilities audited presented accessibility features for people with visual or hearing impairments [19]. Martins and colleagues undertook an objective assessment of the accessibility of Family Health Units, using a structured checklist [13]. They assessed 90 buildings and noted that fewer than half of facilities had a ramp (48%), and hand-rails and anti-slip flooring were also lacking. These findings were similar to those of a physical accessible audit among 147 health units in Ceara State [16]. De Franca and colleagues also highlighted difficulties in access to health units, noting that the greater issues were internal, including lack of accessible drinking facilities and restrooms [15]. Accessibility issues were also highlighted at the hospital level in Brazil. A checklist-based audit of four hospitals in Ceara State showed some successes in terms of presence of ramps and accessible counters, but large gaps such as the lack of accessible drinking fountains and obstacles in internal areas that would impede movement [14]. Accessibility can be assessed through interviews with persons with disabilities, as well as through formal audits with a checklist. Here too, important issues have been highlighted in Brazil. Amaral and colleagues interviewed 523 people with disabilities [12], and many of these reported difficulties in physical accessibility of healthcare facilities (64%) and that there was a lack of special facilities in the healthcare centres (42%). These issues were also highlighted for dental public services, where 37% of patents and 44% of dentists reported inadequate physical access infrastructure (e.g., doors, hallways, waiting rooms and offices) [17].

Studies from other LMICs are sparse but are consistent with our findings for Brazil. A physical accessibility audit was undertaken in a district in Southern India [11]. Externally, most of the primary health care units were considered to be accessible. However, this worsened once the disabled person entered the building, as only one third of the units had accessible doors, and none had adjustable examination tables or accessible toilets. As another example, a study was undertaken on access to primary healthcare for persons with spinal cord injuries in the Gaborone area of Botswana [10]. There were frequent reports of physical inaccessibility, with almost half of people surveyed reporting that they were unable to enter the facility themselves. Our data also tallies with the evidence from high-income settings. For instance, in the USA a number of research teams have used objective site assessments to consider the accessibility of healthcare facilities for people with disabilities. One study of 2,389 primary care facilities in California used a 55-item instrument for site assessments [7]. Although they found that the exterior, interior and building access generally met criteria, there were major issues with lack of appropriate equipment (accessible weight scale, height adjustable examination table). Other studies in the USA paint a bleaker picture, with few primary care clinics meeting standard accessibility requirements [8], and widespread issues in access reported by people with disabilities [9].

Ensuring the accessibility of healthcare facilities for people with disabilities is non-negotiable, enshrined in national and international law [6]. The UNCRPD specifically considers accessibility (Article 9), specifying the appropriate measures that State Partners need to make, including promotion of physical accessibility and accessibility of information. Within countries, there are usually legal requirements that healthcare services are accessible for people with disabilities, and lawsuits may arise if facilities do not meet these standards [28]. However, our data and the existing literature shows that there are large gaps in accessibility, even in Brazil where there is a strong policy commitment to the provision of healthcare for people with disabilities. Improving accessibility of healthcare facilities will ensure that the healthcare system is better able to serve people with disabilities but will also make it more accessible for all. For instance, ramps will benefit not only people with mobility impairments, but also parents with strollers or people who are unwell and have difficulties walking. Provision of information in plain language will be helpful for people with cognitive impairments, but also for those who do not speak the prevailing language well. Furthermore, improving access will increase uptake of healthcare and so produce cost savings later down the line, by avoiding expensive health sequalae. It can also be argued that it will be difficult to achieve Universal Health Coverage, or the Sustainable Development Goals on Health, without provision of healthcare to this large and vulnerable group, and so a focus on accessibility is essential [29,30]. There are therefore a range of reasons for improving accessibility of healthcare facilities.

There are costs to improving accessibility, but these will be minimised if we ensure that facilities are accessible at the design-phase, as retrofitting can be expensive. Accessibility standards themselves may need adjustment as they are often a minimum of what needs to be achieved, rather than what is necessary to achieve true access. Besides financial costs, advocacy and educational investments are needed towards health care professionals, policy makers, managers and society as a whole, in order to sustain a people centred paradigmatic shift in health care. The right to health to people with disabilities implies that services respond to peoples’ needs, covering both the geographic and social organizational dimensions of accessibility [19]. Ideally, we should be aiming for universal design where the healthcare system is appropriate for the full diversity of people with disabilities. Overall, more guidance would be helpful on how to make services, including healthcare, accessible for people with disabilities [31]. Human-centred design approaches should be promoted, whereby facilities are specifically designed to be accessible for the groups most in need–such as people with disabilities [32]. Our study also shows the value of routinely undertaking accessibility audits to highlight issues and using this information to take the health system to account. Making accessibility audits simpler, and mandatory, would be an important step in improving the accessibility of healthcare facilities.

There are important strengths and limitations to our study, which must be considered when interpreting the findings. This study is the first example of a national comprehensive assessment anywhere in the world, including all public primary health care facilities in Brazil. The large scale allows consideration of variation in accessibility by region, municipality size and facility type. A broad assessment of accessibility was taken, incorporating physical accessibility of the internal and external areas, but also consideration of other features of accessibility. Furthermore, a standard objective checklist-tool was used across all facilities, and this was undertaken by trained staff. By contrast, previous assessments have been far smaller, generally more focused on physical accessibility only, and more geographically-focused. However, reliability and validity data were not available for the scales used. All primary healthcare units were included, and so a precise estimate of accessibility was achieved, with narrow confidence intervals. There are also limitations. The audit was undertaken in 2012, and so may not fully reflect today’s situation. The Brazilian law on the Inclusion of Persons with Disabilities was passed in 2015 and so there may have been further initiatives to improve accessibility since the audit. Each accessibility feature was scored as yes or no, yet there may be variation in accessibility (e.g., regular floor in some areas but not others) and so nuances would be missed. Data was collected by researchers from 45 universities, which may have led to variations in reporting. Furthermore, some key aspects of accessibility were not assessed, including the accessibility of equipment. Accessibility of transport to the healthcare facility was not included, even though this presents an important barrier for many people with disabilities [3,4]. The assessment included public (SUS) facilities, but not private facilities. Moreover, the tool was an objective checklist and complementary qualitative data from people with disabilities and healthcare workers would help in our understanding of accessibility issues in Brazil.

## 5. Conclusions

This national assessment of the accessibility of healthcare facilities in Brazil shows that it is feasible to undertake these audits on a large scale and these audits should be repeated in other settings. It highlights important gaps in accessibility, increasing the risk of the violation of the right to health of people with disabilities. The assessment also uncovered structural socioeconomic inequalities among regions and cities in the country which reinforces those gaps. Future work should focus on updating accessibility audits in Brazil and undertaking large scale audits in other settings. There is also a need to develop and trial approaches to overcoming the gaps identified in the audits, whether through incentives or penalties (e.g., fines).

## Figures and Tables

**Table 1 ijerph-18-02953-t001:** Assessment criteria in accessibility audit.

Sub-Scale of Accessibility	Question (Answer, Yes = 1, No = 0)
External	The Health Unit entrance sidewalk is in good condition, that is, it has regular floor, without gaps or holes, with easy displacement for wheelchair users, people with special needs and wheelchair users?
	Does the health facility have a rug?
	Does the health facility have a non-slip floor?
	Does the health facility have a regular floor?
	Does the health facility have a smooth floor?
	Does the health facility have access ramp?
	Does the health facility have handrail?
	Does the health facility have wheelchair-accessible door and entrance corridor?
Internal	Does the health facility have adapted restrooms with higher toilet, sink accessories, lower level soap and paper dispenser, grab bars, door opening out and manoeuvre area that allows wheelchair circulate?
	Does the health facility have grab bars?
	Does the health facility have handrail?
	Does the health facility have wheelchair-accessible interior corridors and doors?
	Does the health facility have interior doors adapted for wheelchairs?
	Does the health facility have space for wheelchair accommodation in the waiting and reception room?
	Does the health facility have adapted drinking fountains?
	Does the health facility have wheelchair available for user travel?
Information	Does the health facility use international symbols for people with physical, visual and hearing disabilities?
	Does the health facility use signage through texts, drawings, colours or figures (visual) that indicate the environments of the Health Unit and the services offered?
	Does the health facility use embossed characters, Braille or embossed figures (tactile)?
	Does the health facility use hearing aids (sound)?
	Does the health facility have professionals to host people with disability?

**Table 2 ijerph-18-02953-t002:** Comparison of five regions in Brazil in terms of demographics, number and distribution of primary health care facilities in 2012.

Characteristics		Southeast (Least Poor)	South	Northeast	Central-West	North (Most Poor)	Total Brazil
Population size 2013 in thousands–*n* (%)		84,465 (42%)	28,795 (14%)	55,795 (28%)	14,993 (8%)	16,983 (8%)	201,033
City Size	Major	6 (35%)	2 (12%)	5 (29%)	2 (23%)	2 (12%)	17
	Large	135 (48%)	50 (18%)	55 (17%)	17 (6%)	24 (8%)	281
	Medium	107 (31%)	52 (15%)	120 (35%)	19 (6%)	42 (12%)	340
	Small	1420 (29%)	1087 (22%)	1614 (33%)	429 (8%)	382 (8%)	4932
Number of PHC facilities		11,943 (31%)	6315 (16%)	14,638 (38%)	2706 (7%)	3210 (8%)	38,812
PHC facility workforce size		195,420 (36%)	85,744 (16%)	184,746 (34%)	39,901 (7%)	43,483 (8%)	549,294
PHC facilities by city size	Major	1012 (45%)	307 (14%)	424 (19%)	229 (10%)	298 (13%)	2270
	Large	3564 (44%)	1309 (16%)	2074 (26%)	483 (6%)	682 (8%)	8112
	Medium	1440 (28%)	730 (14%)	2159 (42%)	297 (6%)	553 (11%)	5179
	Small	5927 (25%)	3969 (17%)	9981 (43%)	1697 (7%)	1671 (7%)	23,251
Graduated professional available at PHU	Yes	8659 (35%)	4048 (16%)	8581 (35%)	1786 (7%)	1654 (7%)	24,728
	No	2315 (22%)	1495 (14%)	4783 (46%)	537 (5%)	1327 (13%)	10,457
	Other	718 (27%)	419 (16%)	1038 (39%)	316 (12%)	198 (7%)	2689
Type of services rendered at PHU	Dentist	7398 (30%)	4606 (19%)	9186 (37%)	1937 (8%)	1526 (6%)	24,653
	Vaccination	8537 (29%)	4649 (16%)	12,117 (41%)	2081 (7%)	1999 (7%)	29,383
	Pharmacy	8064 (28%)	4954 (17%)	11,679 (40%)	1934 (7%)	2597 (7%)	29,228
	Physician	10,491 (31%)	5919 (18%)	11,992 (36%)	2431(7%)	2125 (6%)	32,958
	Dressing	10,868 (31%)	5850 (17%)	13,142 (37%)	2459 (7%)	2751 (8%)	35,070
	Nurse	10,830 (31%)	5668 (16%)	13,208 (38%)	2489 (7%)	2497 (7%)	34,692

**Table 3 ijerph-18-02953-t003:** Accessibility of external area of primary health care facilities, by Region of Brazil and Municipality Size.

		Region					Municipality Size			
Characteristics	Total Brazil	Southeast (Least Poor, Reference)	South	Northeast	Central-West	North (Most Poor)	Major (Reference)	Large	Medium	Small
Entrance floor										
Non-slip	28%	33%	41% *	23%	23% *	16% *	36%	30% *	25% *	28% *
Regular	49%	55%	55%	44% *	48% *	41% *	58%	52% *	48% *	48% *
Smooth	36%	40%	35% *	33% *	40% *	37%	37%	36%	37%	36% *
Access ramp	44%	49%	48%	39% *	43% *	35% *	58%	47% *	41% *	41% *
Handrail at entrance	8%	11%	12%	5% *	5% *	3% *	23%	9% *	6% *	6% *
Accessible door/corridor	35%	44%	46% *	26% *	33% *	20% *	50%	38% *	31% *	33% *
Total external space score (SD)	31.0 (2.0)	35.6 (0.5)	38.1 (0.7) *	26.1 (0.4) *	29.9 (0.9) *	22.7 (0.7) *	40.6 (1.1)	33.1 (0.5) *	29.0 (0.6) *	29.8 (0.3) *

* *p* < 0.05 compared to reference group (bi-tailed).

**Table 4 ijerph-18-02953-t004:** Accessibility of internal space of primary health care facilities, by Region of Brazil and Municipality Size.

		Region					Municipality Size			
Characteristics	Total Brazil	Southeast (Least Poor, Reference)	South	Northeast	Central-West	North (Most Poor)	Major (Reference)	Large	Medium	Small
Adapted restrooms	12%	18%	13% *	7% *	12% *	7% *	30%	14% *	9% *	10% *
Grab bars	14%	20%	18% *	9% *	16% *	7% *	31%	17% *	11% *	12% *
Hand-rail	6%	9%	8% *	3% *	5% *	3% *	19%	8% *	4% *	4% *
Corridors wheelchair accessible	24%	32%	32%	16% *	24% *	11% *	35%	26% *	19% *	22% *
Interior doors adapted for wheelchairs	23%	31%	32%	16% *	24% *	11% *	36%	26% *	20% *	22% *
Wheelchair space in reception	29%	36%	39% *	23% *	27% *	14% *	38%	31% *	27% *	28% *
Adapted drinking fountains	9%	13%	8% *	6% *	11% *	7% *	16%	11% *	7% *	8% *
Wheelchair available	34%	50%	50%	17% *	35% *	18% *	55%	43% *	28% *	31% *
Total internal space score (SD)	18.9 (1.9)	26.4 (0.5)	25.2 (0.6) *	12.1 (0.3) *	19.3 (0.9) *	9.5 (0.6) *	32.4 (1.3) *	22.0 (0.6) *	15.6 (0.6) *	17.2 (0.3) *

* *p* < 0.05 compared to reference group (bi-tailed).

**Table 5 ijerph-18-02953-t005:** Accessibility features of primary health care facilities for people with visual or hearing impairments, by Region of Brazil and Municipality Size.

		Region					Municipality Size			
Characteristics	Total Brazil	Southeast (Least Poor, Reference)	South	Northeast	Central-West	North (Most Poor)	Major (Reference)	Large	Medium	Small
International symbols	1%	2%	2% *	1% *	0% *	1% *	4%	2% *	1% *	1% *
Signage	9%	9%	11% *	8%	6% *	7% *	13%	10% *	9% *	8% *
Embossed characters/braille	0.2%	0.3%	0.1%	0.2%	0.1%	0.1%	0.8%	0.2% *	0.2% *	0.2% *
Hearing resources	0.4%	0.7%	0.3% *	0.2% *	0.2% *	0.2% *	1.1%	0.5% *	0.4% *	0.3% *
Professionals to host people with disabilities	21%	27%	25% *	17% *	19% *	12% *	35%	25% *	20% *	19% *
Total sensory accessibility score (SD)	6.3 (1.0)	7.9 (0.2)	7.7 (0.3)	5.2 (0.2) *	5.0 (0.4) *	3.9 (0.3) *	10.6 (0.6)	7.5 (0.3) *	6.2 (0.3) *	5.6 (0.1) *

* *p* < 0.05 compared to reference group (bi-tailed).

**Table 6 ijerph-18-02953-t006:** Poisson regression estimates of the association of total accessibility score with Region, Municipality size and facility type.

	Total Score (SD)	Unadjusted Proportion Ratio Regression Results (PR/95% CI)	Poisson Regression Adjusted for Region (PR/95% CI)	Poisson Regression Adjusted for Region and Municipality Size (PR/95% CI)	Poisson Regression Adjusted for Region, Municipality Size and Facility Type (PR/95% CI)
Region					
SouthEast (least poor)	23.3 ± 0.3	1.9 (1.9–2.0) *		1.9 (1.9–2.0) *	1.7 (1.7–1.8) *
South	23.7 ± 0.4	2.0 (1.9–2.1) *		2.0 (1.9–2.1) *	1.7 (1.6 –1.7) *
North East	14.5 ± 0.2	1.2 (1.2–1.3) *		1.3 (1.2–1.3) *	1.1 (1.0–1.1) *
Central West	18.0 ± 0.6	1.5 (1.4–1.6) *		1.5 (1.4–1.6) *	1.3 (1.2–1.3) *
North (most poor)	12.0 ± 0.4	Reference		Reference	Reference
Municipality size					
Major	27.9 ± 0.8	1.6 (1.5–1.6) *	1.5 (1.5–1.5) *		1.3 (1.3–1.3) *
Large	20.9 ± 0.4	1.2 (1.2–1.2) *	1.1 (1.1–1.1) *		1.0 (1.0–1.1) *
Medium	16.9 ± 0.4	1.0 (0.9–1.0) *	1.0 (1.0–1.0)		1.0 (0.9–1.0) *
Small	17.5 ± 0.2	Reference	Reference		Reference

Type					
Dentist	22.1 ± 0.2	1.7 (1.7–1.8) *	1.7 (1.6–1.7) *	1.6 (1.6–1.7) *	
Vaccination	20.6 ± 0.2	1.6 (1.6–1.7) *	1.7 (1.6–1.7) *	1.6 (1.6–1.7) *	
Pharmacy	19.7 ± 0.2	1.3 (1.2–1.3) *	1.3 (1.3–1.4) *	1.3 (1.3–1.3) *	
Physician	20.2 ± 0.2	2.0 (1.9–2.0) *	1.8 (1.7–1.8) *	1.7 (1.7–1.8) *	
Dressing	19.7 ± 0.2	2.0 (1.9–2.0) *	1.9 (1.8–2.0) *	1.9 (1.8–1.9) *	
Nurse	19.9 ± 0.2	2.2 (2.1–2.3) *	2.1 (2.0–2.2) *	2.1 (2.0–2.1) *	
Average Brazil	18.7 ± 1.3	Reference	Reference	Reference	

PR: Overall Accessibility Ratio, *: *p* < 0.05.

## Data Availability

Data are available on request from the Ministry of Health in Brazil.

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
