# Peer review of "A National Accessibility Audit of Primary Health Care Facilities in Brazil—Are People with Disabilities Being Denied Their Right to Health?"

_ijerph, 2021, doi:10.3390/ijerph18062953_

Round 1

Reviewer 1 Report

Major Comments

  1. What are the research questions? You mentioned a national assessment of the accessibility of primary health care facilities for people with disabilities in Brazil. I understand that this is main objective of this study. You may wish to more elaborate the objectives or research questions. In general, the study often has 2-3 research questions or sub-objectives. In your study, you seem to explore the associations between accessibility, and region, municipality size, and facility type, etc. You, however, did not mention in the introduction.

  1. I am wondering about questionnaire. The answer about each question could be yes or no. Are not there any issues in this regard? For example, “does the health facility have a regular floor?” may have various issues. First, what if the health facility has a regular floor at only one floor? In that case, would the answer be yes?

  1. Which statistical analysis did you do? You seem to use t-test to compare each group to baseline. I think it would be better to perform ANOVA and post hoc test to compare the value among groups. For the regression, it would be easier for the reader to understand the model if you provide the equation.

Minor Comments

  1. Please cite the references for the statement (Line 57 – 61).

Author Response

Major Comments

  1. What are the research questions? You mentioned a national assessment of the accessibility of primary health care facilities for people with disabilities in Brazil. I understand that this is main objective of this study. You may wish to more elaborate the objectives or research questions. In general, the study often has 2-3 research questions or sub-objectives. In your study, you seem to explore the associations between accessibility, and region, municipality size, and facility type, etc. You, however, did not mention in the introduction.

Response: We have clarified in the introduction that “The research questions are: 1) What is the overall accessibility is of primary health care facilities, and 2) How is accessibility related to region, municipality size, and characteristics of the facility (e.g. number of healthcare workers, services offered, poverty level of region).” (line 99)

  1. I am wondering about questionnaire. The answer about each question could be yes or no. Are not there any issues in this regard? For example, “does the health facility have a regular floor?” may have various issues. First, what if the health facility has a regular floor at only one floor? In that case, would the answer be yes?

Response: The questionnaire response was binary (yes/no), yet as the reviewer points out there could be lack of clarity. We have highlighted in the discussion that “Each accessibility feature was scored as yes or no, yet there may be variation in accessibility (e.g. regular floor in some areas but not others) and so nuances would be missed.” Line 556

  1. Which statistical analysis did you do? You seem to use t-test to compare each group to baseline. I think it would be better to perform ANOVA and post hoc test to compare the value among groups. For the regression, it would be easier for the reader to understand the model if you provide the equation.

Response: We used the Tukey’s test for analysis. It differs from ANOVA because ANOVA calculate the difference between groups, but cannot specify where the differences lie on. On the other hand, Tukey test compares the means of all treatments to them mean of every other treatment, based on a studentized range distribution (similar of t test distribution). The regression equation is as follows:

 E[PR| PRR1, PRR2, PRR3] = exp (b0) x  PRR1 x exp (Region) x  PRR2 x exp (Municipality Size) x  PRR3 x exp(Type)

Where :

  • E [PR  | PRR1, PRR2, PRR3] : expected value for overall accessibility given PRR1, PRR2, PRR3
  • PR: Overall Accessibility Ratio
  • b0 : intercept
  • PRR1: Prevalence Ratio for Region
  • PRR2: Prevalence ratio for municipality size
  • PRR3: Prevalence ratio for Primary Health Unit type

This has now been included in the text (Line 160)

Minor Comments

  1. Please cite the references for the statement (Line 57 – 61).

Response: Examples have been given of national laws protecting the right to accessibility of healthcare facilities for people with disabilities. (Line 74)

Reviewer 2 Report

Abstract and Key words: The abstract of the paper needs revision. Why is the study important? What is the main result? Key word; Primary care?

Suggested you could clarify your aim by writing research questions. Mention something about Socioeconomic (least poor to poor)?

Methods: Please clarify; the characteristics of the Region least poor to poor, according to what? City size? Major, large, medium and small, cut off according to what?

In the section of data processing and analysis about the develop total accessibility score, needs references.

Ethic: According to article “Submission to a research ethics committee was not required for this secondary data analysis” what about ethical considerations?
Results: Your results shows narrow 95%CI (table 4). Have you test by taking a random sample from the data and to look at the CI? How about the weightings impact on the estimates? Is it baseline or reference group?
Discussion: Method discussion: What is the limitation of using yes and no questionnaires? References is missing regarding this. What happen when using researchers from 45 universities in collecting data?
Conclusion needs to clarify.

Author Response

Reviewer 2

  1. Abstract and Key words: The abstract of the paper needs revision. Why is the study important? What is the main result? Key word; Primary care?

Response: The abstract has been revised to highlight the importance of the study (Poor accessibility of healthcare facilities is a major barrier for people with disabilities when seeking care. Yet, accessibility is rarely routinely audited – line 11; In conclusion, large-scale accessibility audits are feasible to undertake – line 22) Primary care has been added as a key word

  1. Suggested you could clarify your aim by writing research questions. Mention something about Socioeconomic (least poor to poor)?

Response: The following has been added “The research questions are: 1) What is the overall accessibility is of primary health care facilities, and 2) How is accessibility related to region, municipality size, and characteristics of the facility (e.g. number of healthcare workers, services offered, poverty level of region).” (line 100)

  1. Methods: Please clarify; the characteristics of the Region least poor to poor, according to what? City size? Major, large, medium and small, cut off according to what?

Response: The regions were classified from least poor to most poor on the basis of mean Gross Regional Product (clarified line 152). We have now switched ranking of North and Central-West. Municipality size is classified on the basis of number of inhabitants (clarified line 150).

  1. In the section of data processing and analysis about the develop total accessibility score, needs references.

Response: We are not clear on what additional references are needed, and would be grateful for clarification.

  1. Ethic: According to article “Submission to a research ethics committee was not required for this secondary data analysis” what about ethical considerations?

Response: We have added further clarification by revising this section to read “Submission to a research ethics committee was not required for this secondary analysis of openly-accessible data. Furthermore, data was not included on individuals and was not of a sensitive nature. ” (line 175)

  1. Results: Your results shows narrow 95%CI (table 4). Have you test by taking a random sample from the data and to look at the CI? How about the weightings impact on the estimates? Is it baseline or reference group?

Response: The results have narrow confidence intervals because the sample size was extremely large, including 38,812 health facilities. We have rephrased “baseline” as “reference” in all the tables and footnotes.

  1. Discussion: Method discussion: What is the limitation of using yes and no questionnaires?

Response: There is a statement in the discussion that “Each accessibility feature was scored as yes or no, yet there may be variation in accessibility (e.g. regular floor in some areas but not others) and so nuances would be missed.”  Line 556

  1. What happen when using researchers from 45 universities in collecting data?

Response: We have included the statement in the discussion that “Data was collected by researchers from 45 universities, which may have led to variations in reporting.” Line 556

  1. Conclusion needs to clarify.

Response: We have updated the conclusion to add the following section: “This national assessment of the accessibility of healthcare facilities in Brazil shows the feasibility of undertaking these audits on a large scale and should be repeated in other settings. It highlights important gaps in accessibility, increasing the risk of the violation of the right to health of people with disabilities. The assessment also highlights that structural socioeconomic inequalities among regions and cities in the country reinforces those gaps. Future work should focus on updating accessibility audits in Brazil and undertaking large scale audits in other settings. There is also a need to develop and trial approaches to overcoming the gaps identified in the audits, whether through incentives or penalties (e.g. fines). ” (line 567)

Reviewer 3 Report

The conclusion could be finalized by presenting the prospect for further research or possible solutions to the problems identified during the research.

Author Response

Reviewer 3

  1. The conclusion could be finalized by presenting the prospect for further research or possible solutions to the problems identified during the research.

Response: We have updated the conclusion to add the following section: “This national assessment of the accessibility of healthcare facilities in Brazil shows the feasibility of undertaking these audits on a large scale and should be repeated in other settings. It highlights important gaps in accessibility, increasing the risk of the violation of the right to health of people with disabilities. The assessment also highlights that structural socioeconomic inequalities among regions and cities in the country reinforces those gaps. Future work should focus on updating accessibility audits in Brazil and undertaking large scale audits in other settings. There is also a need to develop and trial approaches to overcoming the gaps identified in the audits, whether through incentives or penalties (e.g. fines). ” (line 567)

Round 2

Reviewer 2 Report

The authors have responded well to the questions, however there are a few

questions left.

Question 3: Methods: Please clarify; the characteristics of the Region least poor to poor, according to what? City size? Major, large, medium and small, cut off according to what? Response: The regions were classified from least poor to most poor on the basis of mean Gross Regional Product (clarified line 152). We have now switched ranking of North and Central-West. Municipality size is classified on the basis of number of inhabitants (clarified line 150). Please write the index down for the Gross Regional Product

Question 4: In the section of data processing and analysis about the develop total accessibility score, needs references. Response: We are not clear on what additional references are needed and would be grateful for clarification.  What kind of method did you use to (line 124 to 129) to calculate total accessibility score, please add references of this method.

Question 6. Results: Your results shows narrow 95%CI (table 4). Have you test by taking a random sample from the data and to look at the CI? How about the weightings impact on the estimates? Is it baseline or reference group? Response: The results have narrow confidence intervals because the sample size was extremely large, including 38,812 health facilities. We have rephrased “baseline” as “reference” in all the tables and footnotes. Please, discuss this in the section of discussion. Interesting to know if there have been any changes in health policies and so on in Brazil since 2012.

Author Response

1. We have now written the index for Gross Regional Product and provided a reference (line 136).

2. The method for creating the total accessibility score is outlined in the paragraph starting line 121. We do not believe that a reference is necessary. 

3. We have stated in the discussion (line 338): "All primary healthcare units were included, and so a precise estimate of accessibility was achieved, with narrow confidence intervals.". We have also noted (line 341): "The Brazilian law on the Inclusion of Persons with Disabilities was passed in 2015 and so there may have been further initiatives to improve accessibility since the audit."